# The Effectiveness of Physical Therapy in Patients with Generalized Joint Hypermobility and Concurrent Temporomandibular Disorders—A Cross-Sectional Study

**DOI:** 10.3390/jcm10173808

**Published:** 2021-08-25

**Authors:** Małgorzata Kulesa-Mrowiecka, Joanna Piech, Tadeusz S. Gaździk

**Affiliations:** 1Department of Physiotherapy, Institute of Physiotherapy, Faculty of Health Sciences, Jagiellonian University Medical College, 12 Michalowskiego Str., 31-143 Krakow, Poland; tadeusz.gazdzik@uj.edu.pl; 2Faculty of Health Sciences, Jagiellonian University Medical College, 12 Michalowskiego Str., 31-143 Krakow, Poland; joanna.byra@doctoral.uj.edu.pl

**Keywords:** generalized joint hypermobility, temporomandibular disorders, myofascial pain, physiotherapy

## Abstract

Temporomandibular disorders (TMD) consist of a group of symptoms such as: pain of temporomandibular joints, masticatory muscles or surrounding tissues, dysfunctions of TMJs’ mobility, and crepitation. The Hypermobility Joint Syndrome (HJS) manifests in the flaccidity of joint structures, an increase in the range of joint motion, and occurs more often in the young and women. The aim of this study was to present the occurrence of HJS among patients with myogenic TMD and disc displacement with reduction. The secondary goal was to assess the effectiveness of physiotherapy directed to TMD with coexisting HJS. The study involved 322 patients with symptoms of TMD. HJS was diagnosed using the Beighton Scale, which confirmed its occurrence in 26 cases. 79 subjects (7 males and 72 females; mean age, 33.9 ± 10.4 years) were selected and divided into two groups: HJS + TMD (*n* = 26; 2 males and 24 females; mean age, 27.1 ± 9.4 years) and TMD (*n* = 53; 5 males and 48 females; mean age, 37.4 ± 9.2 years). These patients completed 3-week physiotherapy management. Before and after physiotherapy, the myofascial pain severity on Numeric Pain Rating Scale, linear measurement of maximum mouth opening, and opening pattern, were assessed. To demonstrate differences between the results, the level of significance for statistical analysis was set at α = 0.05. A statistically significant improvement was obtained in decreasing myofascial pain in both groups. Coordination of mandibular movements was achieved in both groups. Generalized joint hypermobility occurred among patients with TMD. Physiotherapy directed to TMD was effective in reducing myofascial pain and restoring TMJ’s coordination also in patients with HJS.

## 1. Introduction

Temporomandibular disorders (TMD) affect temporomandibular joints’ (TMJs) structures, masticatory muscles, and surrounding tissues [1]. Symptoms characteristic of these dysfunctions are impaired TMJ mobility and surrounding tissue pain, limitation of joint function, and crepitations [2]. Symptoms are more common in women than men and increase with age [3]. The etiology of functional disorders of the masticatory apparatus is multifactorial and associated with pathologies of the biomechanics of the joint, malocclusions, parafunctions, environmental factors, stress, and hormonal disbalance [4,5,6,7].

Hypermobility Joint Syndrome (HJS) is a generalized, congenital connective tissue disorder caused by a defect in the collagen ratio and occurs with a frequency of 2–57% [8]. Generalized joint hypermobility is almost twice as common in women and dominates in the young. It manifests in the flaccidity of joint structures, the joint capsule and ligaments, and an increase in the range of joint motion compared to the physiological norm, which predisposes the patient to an increased risk of injury (sprains, ligament injuries, joint instability, degenerative changes) [9]. Hypermobility may not show symptoms for years, especially in trained people, athletes, where litheness and flexibility are an asset [10]. However, over time, due to lack of or incorrectly performed activities, overloads of locomotive organ structures may occur, which are manifested in the spine, peripheral joints and muscle pain [11]. Diagnostic criteria for polyarticular flaccidity are based on the Beighton Scale, which measures the performance of five activities bilaterally on a scale of 0 to 1. Generalized hypermobility is found when the patient has received at least 4 points on this scale [12].

HJS could be one of the predisposing factors for TMD [13]. Increasing the TMJs’ range of motion and excessive flaccidity of their joint and periarticular structures leads to early degenerative changes, pain, and inflammation of this joint. Other accompanying symptoms are joint dislocations, joint clicks, dislocation of the articular disc and orofacial pain [14,15,16]. In hypermobility of TMJs, masticatory muscle activity is reduced, and as a result, the chewing process is also disturbed [17]. The assessment of the maximum mouth opening range is an important diagnostic factor for TMJ flaccidity along with the assessment of other movements (protrusion, laterotrusion, mediotrusion) [18].

Physiotherapy, which is a noninvasive method of conservative treatment including manual therapy, exercises, and physical procedures, is used in the therapy of TMD as well as in polyarticular flaccidity. It has been proved that rehabilitation focused on TMD is an essential element of treatment leading to a reduction of pain, improvement of the TMJ’s functions and quality of life [19]. In HJS, physiotherapy is one of the treatments that can constitute the prevention of pain and degenerative changes in the musculoskeletal system [20].

Despite numerous reports on HJS and TMD, there are still no clear results assessing their coexistence and causes, suggesting that further research on this issue is necessary. Therefore, the aim of this study was to present the prevalence of HJS in patients with TMD and to assess the effectiveness of physiotherapy directed to TMD with coexisting HJS in these patients.

## 2. Materials and Methods

A cross-sectional study of the prevalence of HJS among patients with TMD was conducted. Additionally, the clinical effectiveness of physiotherapy in these patients was managed. The study plan and accomplishment were in accordance with the directives of the Declaration of Helsinki. The study was approved by the Ethics Committee of the Jagiellonian University Medical College (ID KBE- 1072.6120.269.2018; date of approval: 25 October 2018). Patients gave their deliberate decision to participate in this study. Stomatognathic physiotherapy was undertaken by a physiotherapist with over 20 years of experience.

### 2.1. Participants

The study and participant recruitment were conducted in the Stomatognathic Clinic in Cracow, Poland. The cross-sectional part of the research included 322 patients (271 women and 51 men, mean age 33.7 ± 13.4 years old) who reported to a physiotherapist with symptoms of TMD. Afterward, 79 subjects (7 males and 72 females; mean age, 33.9 ± 10.4 years) took part in the clinical part of this study assessing the effectiveness of TMD physiotherapy in patients with HJS and TMD. These patients were divided into two groups: HJS + TMD (*n* = 26; 2 males and 24 females; mean age, 27.1 ± 9.4 years) and TMD (*n* = 53; 5 males and 48 females; mean age, 37.4 ± 9.2 years). Patients fulfilling all the following inclusion criteria were eligible for the study group: adults between 18 and 59 years old, women and men, with symptoms of TMD, with diagnosed generalized joint hypermobility confirmed by the Beighton Scale, and deliberate decision to participate in the study. Control group consisted of patients with TMD without HJS. Participants exhibiting at least one of the following criteria were excluded from the study: age under 18 years old and above 59 years old, lack of symptoms of TMD and HJS, acute pain, inflammatory and rheumatic diseases, diseases of the central and peripheral nervous system, dental or physiotherapeutic treatment of the TMD in the last 3 months, pregnancy, and no consent to participate in the study. After undergoing initial screening and meeting the eligibility criteria, individuals who agreed to participate and provided informed consent were included in the study. The allocation and reasons for exclusion are presented in Figure 1.

### 2.2. Intervention

Initially, all patients were examined and diagnosed with TMD according to the Diagnostic Criteria for Temporomandibular Disorders (DC/TMD) [2]. Afterward, patients were asked about the symptoms of HJS and assessed on the Beighton scale.

The Beighton [21] scale is used to assess the symptoms of joint hypermobility and consists of an assessment of 9 movements: (a) passive hyperextension and dorsiflexion of the fifth metacarpophalangeal joint beyond 90° bilaterally, (b) passive flexion of the thumb to touch the forearm bilaterally, (c) passive hyperextension of the knee more than 10° bilaterally, (d) passive hyperextension of the elbow more than 10° bilaterally, and (e) active trunk flexion to place hands flat on the floor without bending knees. Several researchers appoint a score of 0–3 as normal and a score of 4–9 as representing ligamentous laxity [22]; therefore, patients who obtained more than 4 points on the Beighton scale were included in the group of people with hypermobility syndrome.

Physiotherapeutic management focused on the type of TMD for each patient. The main purpose of physiotherapy was to reduce myofascial pain and restore coordination of the mandible. Therapy included manual therapy, postisometric relaxation techniques, trigger point therapy, fascia relaxation techniques over the masticatory muscles, stretching the neck muscles combined with trigger point therapy, exercises of the mandible’s movement coordination, which included strengthening exercises and isometric exercises for masticatory muscles, patient education, and home physical exercises. Patients participated in three physiotherapeutic sessions of 60 min each for three weeks and performed daily home exercises. The examination and therapy of all patients were conducted by one physiotherapist experienced in stomatognathic diagnostics and physiotherapy (more than 20 years of experience).

### 2.3. Outcome Measures

To assess the effectiveness of physiotherapy in the subjects, a functional evaluation of the TMJs was performed, including (a) the linear measurement of the maximum mouth opening, (b) the pain-free maximum mouth opening pattern, and (c) the assessment of pain severity by the Numeric Pain Rating Scale (NPRS) during palpation of masticatory and neck muscles. The assessment was performed twice: during the first session and after 3-week physiotherapy.

A millimeter ruler was used to analyze mandibular movements. The maximum mouth opening was measured along the mandibular midline from the edges of lower incisors to the edges of upper incisors until the first pain was being felt. The distance of the pain-free range of mandible opening movement was reported. The physiological norm of maximum mouth opening was based on the 3-fingers assessment [23], which is a reliable, quick, and simple method of evaluating normal maximum mouth opening. The individual assessment of the ability to position three fingers (index, middle and ring) placed vertically between the upper and lower central incisors up to the distal interphalangeal folds was performed. The distance at the level of the first distal interphalangeal joints was measured by a millimeter ruler. Increased maximum mouth opening was detected when the difference between the tested maximum mouth opening and the individual physiological norm of mouth opening was more than 5 mm.

Additionally, the assessment of the opening pattern was performed, including the occurrence of the mandibular deviation or deflection, to identify a lack of coordination in the movements of the mandibular condyle. A millimeter ruler was placed on the midline between the upper and lower incisors centrally to assess the path of mandibular deviation during the maximum mouth opening three times. The patient’s result was then recorded on a movement diagram (straight line illustration for no deviation, either an oblique line to the left or the right refers to the lateral deviation of the lower jaw, which is perceptible to the right or left at maximum opening). The measurements were performed in a relaxed lying position.

The clinical examination of the selected superficial masticatory and neck muscles consisted of a visual assessment of their structure, shape, and function during mandibular movements and palpation. The muscles were palpated with pressure applied at the muscle attachments and in the thickest area of the muscle fibers. The temporal muscles were examined simultaneously on two sides of the face, considering the three parts of the muscle (anterior, medial, and posterior). The masseter muscles were examined using an intraoral grip: the anterior part of the masseter muscle with the position of the fingers at the beginning of the muscle 1 cm anteriorly from the TMJ, just below the zygomatic arch; the medial part of the masseter muscle below the zygomatic process in the front of the muscle; and the posterior part of the masseter muscle 1 cm above and forward from the angle of the mandible [2]. Because the source of the pain of the masticatory muscles could have been cervical muscles; the descending part of the trapezius muscle and the sternocleidomastoid muscle were also palpated, bilaterally. The trapezius muscle was examined at the upper attachment below the external occipital protuberance, the middle part of the muscle in the largest cross-section area, and the lower part at the shoulder end of the clavicle. The sternocleidomastoid muscle was tested midway through the thickest cross-section of muscle fibers. During each compression, the study participants were asked to rate the level of myofascial pain on the NPRS. The verbal assessment was conducted in an 11-point NPRS, where 0 was no pain and 10 was the most severe pain. The palpation was performed in a relaxed lying position, with support to the neck and back.

### 2.4. Statistical Analysis

The results were subjected to statistical analysis using STATISTICA 13.3 (TIBCO Software Inc., Palo Alto, CA, USA). The basic characteristics were presented using the mean, standard deviation, and percentage. The data were analyzed for compliance with normal distribution using the Shapiro-Wilk test. A paired *t* test was used for within-group comparison of the maximum mouth opening. A Wilcoxon signed rank test was used for within-group comparison of the myofascial pain. An independent *t* test and Mann-Whitney U was used for comparison of differences between the groups. The significance level was set at α = 0.05 for all the analyses.

## 3. Results

The HJS was confirmed in 26 of the 322 participants with TMD (*n* = 26, 8.1%). An independent *t* test and Mann-Whitney U test was used. A total of 79 subjects (7 males and 72 females) participated in the clinical part of the study. The baseline characteristics of the subjects and TMD symptoms are presented in Table 1 and Table 2. There was a statistically significant difference between the two groups in age (*p* < 0.001). In the control group, more participants were older, which may also affect the distribution of education in this group (more patients with higher education).

Among these patients, the most dominant signs and symptoms of TMD were myofascial pain, TMJ sounds like clicking, and deviation in the opening pattern. Myogenic TMD and combined myogenic TMD with disc displacement with reduction were common in both groups. In the study group, disc displacement with reduction was found in 61.5% of patients (*n* = 16) and co-occurred with myofascial pain. The remaining 10 participants (38.5%) had a myogenic TMD.

The assessment of the TMJs’ movement was performed using a linear measurement during active mouth opening with an evaluation of the mandibular movement pattern. Among patients diagnosed with HJS, the maximum mouth opening was normal (*n* = 15, 57.7%) or increased (*n* = 11, 42.3%) compared to the individual physiological norm. Patients in the HJS group were characterized by an increased range of mouth opening, while in the control group the main problem was the limitation of mouth opening.

In the HJS group, deviation-type mandibular movement dysfunction (*n* = 19, 73.1%) was similar to the control group (*n* = 37, 69.8%). Disturbances in the coordination of the mouth opening movement consisted of a deviation-type mandibular movement dysfunction deviating from the midline to one side during mouth opening and returning to the center in the final range of mouth opening.

The level of pain was assessed during palpation of reference points for selected muscles, i.e., temporal, masseter, sternocleidomastoid (SCM), and trapezius, examined bilaterally. The analysis of pain sensation in the NPRS showed that myofascial pain was a common symptom in the study and control groups (all participants).

As a result of physiotherapy, the maximum mouth opening significantly increased in both groups, which was confirmed by the paired *t* test. (*p* < 00.1) (Table 3).

As a result of the therapy, similar visual improvement in mandibular coordination was obtained in 10 out of 19 patients in the HJS group and in 26 out of 37 patients in the control group (Table 4).

Physiotherapy decreased myofascial pain in all assessed muscles, and it was statistically significant in both groups (Wilcoxon signed-rank test *p* < 0.05) (Table 5).

## 4. Discussion

The aim of this study was to assess the prevalence of generalized joint hypermobility among patients with TMD. An additional purpose was the evaluation of the effectiveness of physiotherapy focused on TMD in these patients. The co-occurrence of these two symptoms was confirmed in 26 participants (8.1%), and the main symptoms of TMD in these patients were: increased range of motion, myofascial pain in masticatory and neck muscles, articular disc-related dysfunctions like clicking in TMJs, and disturbances in mandible movement coordination.

The study group consisted of mostly women (*n* = 24, 92.3%) because, as reported by numerous studies on hypermobility, it most often affects females, and this is associated with endocrine disturbances. Women have more flexible joint structures than men, due to the incidence of hypermobility being higher in this sex group [24]. The studies evaluating the mobility of the knee during the menstrual cycle suggest that taking hormone replacement therapy or increased estrogen hormones promotes an increase in the mobility of the knee [25]. However, testosterone has the opposite effect on the joint [26]. The relationship between the increased incidence of TMD in women is not fully understood. It has been noticed that TMD concerns mainly women, and may be associated with hormonal changes [27,28,29].

The prevalence of TMD is also related to age and is most common in people between the ages of 20 and 40 [30]. On the other hand, HJS also occurs mainly in young people [8]. The above reports are consistent with our own observations; the group of patients with the occurrence of symptoms of HJS and TMD was younger than the general group of people who had symptoms of TMD only without HJS.

The presence of HJS has been shown to be a risk factor for the development of TMD [14,18,31,32]. An epidemiological study in Taiwan’s population by Chang et al. [33] demonstrated an association between TMD and HJS, where among people with symptoms of temporomandibular disorders, HJS occurred at a rate of 3.85%. However, studies based on the search for HJS symptoms among people diagnosed with TMD showed a significantly higher rate of HJS prevalence, from 51% [34] to even 79.7% [32]. An observation of the study indicates that HJS was a common disorder among patients with TMD with a prevalence of 8.1%. The above difference may be caused by the larger size of the study group in this study (*n* = 322) than in the cited studies [18,32]. Additional factors contributing to these differences may be socioeconomic and ethnic factors.

The last literature review by Dijkstra et al. [31] suggests that despite the undertaken research, it is still unclear whether there is a relationship between the occurrence of generalized joint hypermobility and TMD, although the literature suggests that hypermobility may be a predisposing factor for the occurrence of TMD.

HJS leads to an increase in the range of motion of multiple joints [9]. The temporomandibular joints can also be exposed to excessive mobility, which most often leads to their instability, subluxation, or disc-related problems [2]. However, there is still no clear influence of HJS on the mobility of the temporomandibular joints. Chiodelli et al. [35] proved that hypermobility does not influence TMJs’ range of motion, but they point out that HJS may be associated with an increased percentage of malocclusion, especially in the case of cross-bite. Similar reports by Barrera-Mora et al. [36] confirmed that there is no relationship between HJS and TMD, but they showed that there is a correlation between the occurrence of malocclusion among patients with HJS, especially in the case of open bite and malocclusion Angle class II. Also, Akgöl et al. [37] indicated that there is no relationship between the maximum mouth opening and HJS, and also no relationship between TMD and HJS. However, there are also contrary reports that hypermobility patients had an increased range of motion of the maximum mouth opening, and more likely painful mouth opening [18], and also lower risk of limited mouth opening [14]. In the current study, the maximum mouth opening was normal in 57.7% of patients and increased in 42.3% of them. Despite conflicting reports on the effect of HJS on TMJ mobility, it should be considered that the occurrence of increased TMJ mobility may be one of the symptoms of HJS, and patients should be assessed for the presence of other HJS symptoms.

TMD may affect the temporomandibular joint disc. Symptoms of articular disc problems are TMJ sounds like clicking, articular disc dislocation, and mandibular deviation [2]. The disc displacement with reduction presented in 61.5% of patients with HJS (*n* = 16) compared to patients without HJS (*n* = 10) and were related to the mandible’s open pattern deviation (*n* = 19) and clicking sounds (*n* = 19). In a cross-sectional study on HJS and TMD, Hirsch et al. [14] demonstrated that subjects who suffer from HJS had a greater risk of suffering nonpainful reproducible reciprocal clicking and had a higher risk of disc displacement with reduction. Despite the lack of a relationship between TMD and HJS, Chang et al. [33] also suggested that the most common form of TMD among patients with hypermobility is disc-related dysfunctions. Nevertheless, Sáez-Yuguero et al. [34] examined a group of women with MRI-confirmed symptoms of TMJ disc dysfunction for the presence of HJS and did not notice a significant statistical relationship between these two dysfunctions. However, Chiodelli et al. [35] noticed a higher percentage of uncorrected lateral deviation during mouth opening and articular noises in patients with hypermobility, but without statistically significant differences. Among the reports, there is no consensus regarding the impact of HJS on articular disc problems in TMD patients.

Another symptom of TMD is pain, which can involve TMJs, masticatory and/or neck muscles, or surrounding tissues. Pain may occur during movement, palpation, or rest. The analysis of pain sensation in the NPRS showed that myofascial pain during palpation of masticatory and neck muscles was a common symptom among patients with HJS and TMD (all participants). Sáez-Yuguero et al. [34] noted that the most common reason for seeking help in treating TMD was pain, which occurred in both positive and negative hypermobility individuals. Akgöl et al. [37] showed a relationship between pain in the temporal muscle at night and the occurrence of congenital polyarticular laxity. However, palpation of the masseter muscle presented no greater pain in hypermobility and TMD patients compared to nonhypermobile TMD patients. During chewing, more pain occurred in people without hypermobility. Davoudi et al. [17] assessed the activity of the masticatory muscles during maximum contraction and chewing in people with hypermobility of the TMJs. Muscle activity in EMG has been shown to be reduced in people with excessive TMJ range of motion. Additionally, in severe cases of mandibular hypermobility, chewing had a lower efficacy compared to healthy subjects. In contrast, studies by Pasinato et al. [18] indicated a relationship between hypermobility and the occurrence of both painful and painless TMD, whereas Hirsch et al. [14] suggested that HJS is not related to myalgia in TMD, but rather to the painless form of TMD.

However, the goal of physiotherapy in the two groups differed; the purpose in the HJS group was not to increase the range of the mouth opening, but to observe how the proposed therapy affects the range of mouth opening. It is likely that the relaxation exercises aimed at reducing myofascial pain resulted in the elongation of the muscle structures and caused a slight increase in maximum mouth opening. On the other hand, in the control group, there was an increase in mouth opening, which was consistent with the objectives of therapy for this group, which mainly had a limited range of mouth opening.

The present study pays attention to the need for a holistic approach to patients with hypermobility and TMD. However, despite all efforts, this study has some limitations. The first limitation is related to the small size of the study group, which translates into the inability to generalize the results to the population of people with HJS. Secondly, for financial reasons, many participants completed therapy after 3 weeks, which prevented further observation of achieved therapy effects, and it was a relatively short intervention time. The third limitation was the lack of homogeneity of groups in age and TMD diagnosis, which could be a potential confounding factor. Future studies are necessary for investigating the long-term effects of physiotherapy in larger and homogeneous TMD patients with HJS.

## 5. Conclusions

To sum up, TMD occurs with generalized joint hypermobility and is related to increased TMJ range of motion, myofascial pain in masticatory and neck muscles, articular disc-related dysfunctions like clicking in TMJs, and disturbances in mandible movement coordination. Physiotherapy focused on TMD in patients with HJS is effective in reducing pain and improving the coordination of mandible movements.

## Figures and Tables

**Figure 1 jcm-10-03808-f001:**
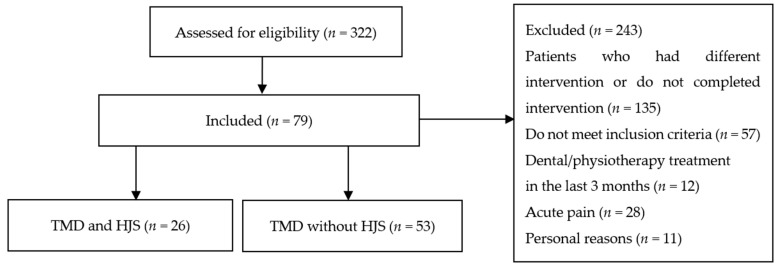
Participants allocation to the clinical part of the study. HJS—Hypermobility Joint Syndrome; TMD—Temporomandibular Disorders.

**Table 1 jcm-10-03808-t001:** Baseline characteristics of the participants.

Variables	Study Group		Control Group		*p*-Value
*n*	26		53		
Age (mean, sd)	27.1 ± 9.4		37.4 ± 9.2		<0.001 ^1^
Gender (*n*, %)					
Male	2	7.7%	5	3.8%	
Female	24	92.3%	48	96.2%	
Education (*n*, %)					
Higher	14	53.8%	44	76.9%	
Secondary	12	46.2%	9	23.1%	

^1^ Independent sample *t*-test.

**Table 2 jcm-10-03808-t002:** Baseline characteristics of TMD symptoms among participants.

Variables	Study Group		Control Group		*p*-Value
Type of TMD * (*n*, %)					
Myofascial pain	26	100%	53	100%	
Myogenic TMD + Disc displacement with reduction	16	61.5%	10	18.9%	
Maximum mouth opening [mm] (mean, sd)					
	47.9 ± 5.5		42.1 ± 6.9		0.0002 ^1^
Maximum mouth opening (*n*, %)					
normal	15	57.7%	21	39.6%	
increased	11	42.3%	4	7.5%	
decreased	0	0%	28	52.9%	
Deviation of the mandible (*n*, %)					
to the right	11	42.3%	17	32.1%	
to the left	8	30.8%	20	37.7%	
without deviation	7	26.9%	16	30.2%	
TMJs ** sounds (clicking)	19	73.1%	27	50.9%	
Myofascial pain [NPRS] (mean, sd)					
Masseter	9.1 ± 1.9		9.1 ± 1.2		0.523 ^2^
Temporal	5.8 ± 2.8		3.9 ± 2.1		0.0008 ^2^
Sternocleidomastoid	8.7 ± 1.7		7.5 ± 2.6		0.155 ^2^
Trapezius	8.5 ± 2.2		8.6 ± 1.9		0.88 ^2^

* TMD—temporomandibular disorders; ** TMJs—temporomandibular joints, NPRS—Numeric Pain Rating Scale; ^1^ Independent *t* test, *p* < 0.05; ^2^ Mann–Whitney U test.

**Table 3 jcm-10-03808-t003:** Changes in the maximum mouth opening at pre- and post-intervention (*n* = 52).

	Intervention	HJS ** + TMD ***	TMD	*p*-Value ^2^
MMO * [millimeters]	Pre	47.9 ± 5.5	42.1 ± 6.9	0.0002
	Post	50.8 ± 4.4	48.5 ± 5.1	0.001
*p*-value ^1^		0.000003	<0.001	

Values are presented as mean ± standard deviations; * maximum mouth opening; ** hypermobility joint syndrome; *** temporomandibular disorders; ^1^ Paired *t* test, *p* < 0.05; ^2^ Independent *t* test, *p* < 0.05.

**Table 4 jcm-10-03808-t004:** Changes in the deviation of mandibular opening pattern before and after physiotherapy (*n* = 52).

**Deviation of Mandibular Opening Pattern**	**Treatment Group**	***n*** **, %**			
		**before Physiotherapy**		**after Physiotherapy**	
	HJS * + TMD **	19	73.1%	10	52.6%
	TMD	37	69.8%	26	49.1%

* hypermobility joint syndrome; ** temporomandibular disorders.

**Table 5 jcm-10-03808-t005:** Changes in the myofascial pain at pre- and post-intervention (*n* = 52).

Muscle [NPRS *** 0–10]	Intervention	HJS * + TMD **	TMD	*p*-Value ^2^
Masseter	Pre	9.1 ± 1.9	9.1 ± 1.2	0.523
	Post	5.3 ± 1.1	5.9 ± 1.8	0.06
*p*-value ^1^		<0.001	<0.001	
Temporal	Pre	5.8 ± 2.8	3.9 ± 2.1	0.0008
	Post	2.9 ± 1.4	2.0 ± 1.3	0.002
*p*-value ^1^		<0.001	<0.001	
Sternocleidomastoid	Pre	8.7 ± 1.7	7.5 ± 2.6	0.155
	Post	5.0 ± 1.4	4.8 ± 2.4	0.503
*p*-value ^1^		<0.001	<0.001	
Trapezius	Pre	8.5 ± 2.2	8.6 ± 2.0	0.88
	Post	4.2 ± 1.5	5.4 ± 2.0	0.005
*p*-value^1^		<0.001	<0.001	

Values are presented as mean ± standard deviations. * hypermobility joint syndrome. ** temporomandibular disorders. *** NPRS-Numeric Pain Rating Scale. ^1^ Wilcoxon signed rank test, *p* < 0.05. ^2^ Mann-Whitney U test, *p* < 0.05.

## Data Availability

The data presented in this study are available on motivated request to the corresponding author.

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
