# Peer review of "The Effectiveness of Physical Therapy in Patients with Generalized Joint Hypermobility and Concurrent Temporomandibular Disorders—A Cross-Sectional Study"

_jcm, 2021, doi:10.3390/jcm10173808_

Round 1
Reviewer 1 Report
Thanks for doing the requested amendments, the quality of the paper has gained significantly by adding a control group.
The following points as noticed when reading are recommended for checking/modification:
Abstract:
please change : … the statistical analysis was used with a significance level of α=0.05. Put instead e.g.: level of significance for statistical analysis was set at α=0.05.
Please make message clear and put: Physiotherapy directed to TMD is effective in reducing myofascial pain and restoring TMJ’s coordination also in patients with HJS.
Materials and Methods:
Please rephrase in figure1: Patients who had different intervention or do not completed intervention (n=135)
Put instead: Patients who underwent different interventions or did not complete intervention (n=135)
Results.
Please add test: An independent t test and Mann-Whitney U test was used
Please correct: of the subjects and TMD symptoms were presented in Table 1 and Table 2. Put instead: of the subjects and TMD symptoms are presented in Table 1 and Table 2.
Please modify: In the study group disc displacement with reduction was found in 61.5% of patients (n=16) and co-occurred with myofascial pain.
Lines 233 and 234: Disturbances in the coordination of the mouth opening movement consisted in a deviation-type mandibular movement dysfunction
(consisted of would be correct if there were no other disturbances of coordination or if all kinds of disturbances were listed here)
Lines 254-260: please modify , this needs to be transferred into the discussion part: However, the goal of physiotherapy in the two groups differed The purpose in HJS group was not to increase the range of the mouth opening, but to observe how the proposed therapy affects the range of mouth opening. It is likely that the relaxation exercises aimed at reducing myofascial pain resulted in the elongation of the muscle structures and caused a slight increase in maximum mouth opening. On the other hand, in the control group, there was an increase in mouth opening, which was consistent with the objectives of therapy for this group, which mainly had a limited range of mouth opening.
Author Response
Response to Reviewer 1 Comments
Dear Reviewer,
We would like to thank you for reviewing our manuscript. We appreciate all comments and effort put into writing the review. We are thankful for your positive comments and constructive advice that has helped us improve the manuscript. All the changes were introduced using the "Track Changes" function in Microsoft Word, so that changes are easily visible.
The following points we modificated:
Abstract:
“… level of significance for statistical analysis was set at α=0.05.”- changed according to Reviewer suggestions
Changed according to Reviewer suggestions: “Physiotherapy directed to TMD is effective in reducing myofascial pain and restoring TMJ’s coordination also in patients with HJS.”
Materials and Methods:
In figure 1: Patients who underwent different interventions or did not complete intervention (n=135)- Changed according to Reviewer suggestions
Results.
Added according to Reviewer suggestions: “An independent t test and Mann-Whitney U test was used”
Corrected according to Reviewer suggestions : “ of the subjects and TMD symptoms are presented in Table 1 and Table 2.”
Modified according to Reviewer suggestions : “In the study group disc displacement with reduction was found in 61.5% of patients (n=16) and co-occurred with myofascial pain.”
Lines 233 and 234: Disturbances in the coordination of the mouth opening movement consisted in a deviation-type mandibular movement dysfunction- in patients with HJS, we suggest to not change.
We transferred into the discussion part lines 381-387: please modify , this needs to be: However, the goal of physiotherapy in the two groups differed The purpose in HJS group was not to increase the range of the mouth opening, but to observe how the proposed therapy affects the range of mouth opening. It is likely that the relaxation exercises aimed at reducing myofascial pain resulted in the elongation of the muscle structures and caused a slight increase in maximum mouth opening. On the other hand, in the control group, there was an increase in mouth opening, which was consistent with the objectives of therapy for this group, which mainly had a limited range of mouth opening.
Kindly we thank You once again
With best wishes
Yours sincerely- the Authors
Response to Reviewer 1 Comments
Dear Reviewer,
We would like to thank you for reviewing our manuscript. We appreciate all comments and effort put into writing the review. We are thankful for your positive comments and constructive advice that has helped us improve the manuscript. All the changes were introduced using the "Track Changes" function in Microsoft Word, so that changes are easily visible.
The following points we modificated:
Abstract:
“… level of significance for statistical analysis was set at α=0.05.”- changed according to Reviewer suggestions
Changed according to Reviewer suggestions: “Physiotherapy directed to TMD is effective in reducing myofascial pain and restoring TMJ’s coordination also in patients with HJS.”
Materials and Methods:
In figure 1: Patients who underwent different interventions or did not complete intervention (n=135)- Changed according to Reviewer suggestions
Results.
Added according to Reviewer suggestions: “An independent t test and Mann-Whitney U test was used”
Corrected according to Reviewer suggestions : “ of the subjects and TMD symptoms are presented in Table 1 and Table 2.”
Modified according to Reviewer suggestions : “In the study group disc displacement with reduction was found in 61.5% of patients (n=16) and co-occurred with myofascial pain.”
Lines 233 and 234: Disturbances in the coordination of the mouth opening movement consisted in a deviation-type mandibular movement dysfunction- in patients with HJS, we suggest to not change.
We transferred into the discussion part lines 381-387: please modify , this needs to be: However, the goal of physiotherapy in the two groups differed The purpose in HJS group was not to increase the range of the mouth opening, but to observe how the proposed therapy affects the range of mouth opening. It is likely that the relaxation exercises aimed at reducing myofascial pain resulted in the elongation of the muscle structures and caused a slight increase in maximum mouth opening. On the other hand, in the control group, there was an increase in mouth opening, which was consistent with the objectives of therapy for this group, which mainly had a limited range of mouth opening.
Kindly we thank You once again
With best wishes
Yours sincerely- the Authors

Reviewer 2 Report
Movements of mandibular laterality may be affected by meniscal displacement, most commonly towards the medial aspect of the joint. MRI was not considered for the patients in the study ?
Author Response
Response to Reviewer 2 Comments
Dear Reviewer,
We would like to thank you for reviewing our manuscript. We appreciate all comments and effort put into writing the review. We are thankful for your positive comments and constructive advice that has helped us improve the text. All the changes were introduced using the "Track Changes" function in Microsoft Word, so that changes are easily visible.
The followings are our responses:
„Movements of mandibular laterality may be affected by meniscal displacement, most commonly towards the medial aspect of the joint. MRI was not considered for the patients in the study ?”
The exclusion criteria: acute pain, inflammatory and rheumatic diseases, diseases of the central and peripheral nervous system, made that there was not obligatory the MRI for this patients. According Wilkes Classification Internal derangement of TMJ in groups III-V is necessary the MRI. The study group represented I-II stage in Wilkes Classification without ankylosis. In Poland first choice diagnosis procedure is still CT and functional X-ray for TMJ. The MRI is not recommend to every patient with TMD. The patients feel claustrophobic and uncomfortable inside the scanner so this type of diagnosis is recommended for III -V stage in Wilkes Classification.
Kindly we thank You once again
With best wishes
Yours sincerely- the Authors

This manuscript is a resubmission of an earlier submission. The following is a list of the peer review reports and author responses from that submission.
Round 1
Reviewer 1 Report
With “The effectiveness of physical therapy in patients with generalized joint hypermobility and temporomandibular disorders a cross sectional study” the authors address a controversially discussed and so far, rather unresolved topic. Therefore, the subject is well worthy of further investigation. The article is written fluently in apparently good English and - apart from some minor redundancies regarding intermingling results and methods (e.g. lines 194-198 and 202-203) - is well structured. The authors offer a comprehensive overview of the respective literature and the discussion is quite informative (except for some repetition of results, e.g. lines 223-228).
Nevertheless, the article in its present form is flawed by a series of major drawbacks which need to be adequately addressed before consideration for publication.
1st: lack of control group
As the authors point out themselves (lines 316-317), a major limitation is lack of a control group which is an essential to allow for judgement of effectiveness of physiotherapy in HJS with TMD. As this is basically a retrospective study according to the authors’ own statement (cf. line 317), it should be easily feasible to procure respective data from the general group of 322 minus 26 HJS patients; if not available, it should be no bigger problem to do a respective assessment in a non HJS group with myogenic TMD.
2nd: myogenic TMD instead of TMD in general
As TMD is an umbrella term for a complex group of different pathologies (and certainly not a triad as mentioned in the abstract intro lines 12-13, which needs to be changed e.g. quoting lines 31-34, which put it differently, but correctly), the authors should specify the article with regard to myogenic disorders and DDwR (mentioning this early in the abstract ; and please not referring to “orofacial pain”, which is an umbrella term, too, but to myofascial pain instead).
3rd: DDwR and HJS
Please also note that DDwR may be attributed to HJS, but there are no data given in the article supporting this conclusion; at least not unless data of a representative control group are supplied allowing for a comparison; thus, this potentially interesting information remains speculative and is not supported by data (solution Table 1: should contain the data for the overall collective without HJS)
4th: assessment of myofascial pain
The authors give a detailed description of assessment of the respective muscle groups. Nevertheless, with regard to muscle assessment , the established standard in this field is assessment according to the DC/TMD protocol to allow for comparison with published data; as this protocol was not used, a control group thus becomes even more indispensable.
5th: inhomogeneity of the collective:
26 patients with HJS were included, however, 4 out of 26 patients didn’t show muscle pain (Table 2); therefore, the rationale for physiotherapy remains unclear; basically the 4 patients are no TMD patients, unless the lead symptom was joint noise, so without pain no indication for physiotherapy, either: therefore, the collective should run 22 instead of 26, then. Otherwise the authors should address this point of criticism in the rebuttal or in the discussion.
6th: subgroups among HJS & concurrent TMD patients
Next, the myofascial pain group n=22 minus? should be differentiated from the internal derangement (ID) group with DDwR plus again the mixed group pain & ID derangement group t(so there are basically 3 different patient sub-collectives) to allow for sound nosologically based conclusions. Otherwise this means comparing apples and oranges (which comes back to the problem of small numbers problem address in line313)
In this context, with a rate of approximatively 10% among TMD, larger collectives could be realized e.g. by a multicentre design.
7th: assessment of MMO as parameter for effectiveness:
The MMO parameter is highly questionable: does it really make sense to even reinforce mobility of an already hypermobile joint and use this as an indicator for effectiveness of physiotherapy? MMO is rather a surrogate parameter for reduction of muscle pain, so the authors should focus on myofascial pain as target parameter in HJS patients (if present, cf. above), at least this point needs to be addressed in the discussion part.
8th: conclusion part is highly speculative:
a) conclusion can occur with (generalized) HJS (without control group of both a normal population and a non HJS control group) this massage is either trivial (of course they can occur) or unfounded by supplied data;
b) related to increased ROM (cannot be stated without control group and respective statistical evaluation, so conclusion remains speculative)
c) myofascial pain (cf. above, plus no subgroup analysis performed, therefore not based on presented data and thus speculative)
d) articular related dysfunction (i.e. internal derangement Wilkes I & II) : cf. above, not supported due to lack of a control group
e) coordination disturbances: mentioned, but no analysis done, this not supported by data, speculative
f) physiotherapy is effective in reducing pain (yes, but no control without HJS, so less meaningful) and improving the coordination of the mandible (no data supplied, therefore speculative)
Some minor points to be addressed/clarified:
- sequentially assessed NPS is rather subjective evaluation tool – the authors should also state that it was in fact one examiner (with > 20 years of experience) or were there several examiners experienced > 20 years? Should be made clear. If several, interrater assessment would be required (IR variability etc.).
- just to avoid any misunderstanding: the authors mention the term deviation assessed as more than 5 mm abduction from the midline; do the authors mean deflection instead or really deviation? (with the latter one of course making sense as there was no DDwoR in the collective)
- last but not least the reviewer recommends to modify the title of the article to run: “The effectiveness of physical therapy in patients with generalized joint hypermobility and concurrent temporomandibular disorders - a cross sectional study”
Author Response
Dear Reviewer,
We would like to thank you for reviewing our manuscript. We appreciate all comments and effort put into writing the review. We are thankful for your positive comments and constructive advice that has helped us improve the manuscript. All the changes were introduced using the "Track Changes" function in Microsoft Word, so that changes are easily visible.
Kindly we thank You once again
With best wishes
Yours sincerely- the Authors
The followings are our point-by-point responses:
With “The effectiveness of physical therapy in patients with generalized joint hypermobility and temporomandibular disorders a cross sectional study” the authors address a controversially discussed and so far, rather unresolved topic. Therefore, the subject is well worthy of further investigation. The article is written fluently in apparently good English and - apart from some minor redundancies regarding intermingling results and methods (e.g. lines 194-198 and 202-203) - is well structured. The authors offer a comprehensive overview of the respective literature and the discussion is quite informative (except for some repetition of results, e.g. lines 223-228).
Nevertheless, the article in its present form is flawed by a series of major drawbacks which need to be adequately addressed before consideration for publication.
1st: lack of control group
As the authors point out themselves (lines 316-317), a major limitation is lack of a control group which is an essential to allow for judgement of effectiveness of physiotherapy in HJS with TMD. As this is basically a retrospective study according to the authors’ own statement (cf. line 317), it should be easily feasible to procure respective data from the general group of 322 minus 26 HJS patients; if not available, it should be no bigger problem to do a respective assessment in a non HJS group with myogenic TMD.
The study was carried out as an analysis of the medical records of patients who underwent physiotherapy due to dysfunctions in temporomandibular joints. Among them, only patients with TMD and HJS were selected. The lack of a control group was due to the large variation in the remaining data and missing data. It is planned to extend this study to a control group with a repeat of the study protocol. We added the limitations and pointed out strengths - Corrected according to Reviewer suggestions (lines: 330-332).” The third limitation is the lack of a control group due to the retrospective nature of the study. Future studies should include a control group and a long term of follow-up assessment”
Physiotherapy for treating TMD in HJS is less studied in the current literature and requires further research on bigger sample on effectiveness physiotherapy intervention.
2nd: myogenic TMD instead of TMD in general
As TMD is an umbrella term for a complex group of different pathologies (and certainly not a triad as mentioned in the abstract intro lines 12-13, which needs to be changed e.g. quoting lines 31-34, which put it differently, but correctly), the authors should specify the article with regard to myogenic disorders and DDwR (mentioning this early in the abstract ; and please not referring to “orofacial pain”, which is an umbrella term, too, but to myofascial pain instead).
TMD description changed in abstract and orofacial pain term changed into myofascial pain (line 12-13, 16-17, 20, 27)- Corrected according to Reviewer suggestions
3rd: DDwR and HJS
Please also note that DDwR may be attributed to HJS, but there are no data given in the article supporting this conclusion; at least not unless data of a representative control group are supplied allowing for a comparison; thus, this potentially interesting information remains speculative and is not supported by data (solution Table 1: should contain the data for the overall collective without HJS)
The results in the Table 2 and in the lines 212-214 shows that among study group there were 16 patients with disc displacement with reduction and co-occurred myofascial pain.
Table 1. Baseline characteristics of the patients with HJS- only patients with TMD and HJS were selected. The lack of a control group was due to the large variation in the remaining data and missing data. It is planned to extend this study to a control group with a repeat of the study protocol- it was pointed in the discussion.
4th: assessment of myofascial pain
The authors give a detailed description of assessment of the respective muscle groups. Nevertheless, with regard to muscle assessment , the established standard in this field is assessment according to the DC/TMD protocol to allow for comparison with published data; as this protocol was not used, a control group thus becomes even more indispensable.
Muscle assessment was based on the DC/TMD, but only selected muscles were included in the analysis of the effects of physiotherapy on myofascial pain. We considered this limitation as an opportunity to identify new paradigm in physiotherapy and prevention for patients with HJS and highlight the need for further development in this area of study and we plan further research- according to Reviewer suggestions
5th: inhomogeneity of the collective:
26 patients with HJS were included, however, 4 out of 26 patients didn’t show muscle pain (Table 2); therefore, the rationale for physiotherapy remains unclear; basically the 4 patients are no TMD patients, unless the lead symptom was joint noise, so without pain no indication for physiotherapy, either: therefore, the collective should run 22 instead of 26, then. Otherwise the authors should address this point of criticism in the rebuttal or in the discussion.
There is information about myofascial pain that was relevant to all participants: in the discussion (line 304-307) and results (line 211-212). The main reason for physiotherapy was pain. As a result of a mistake, Table 2 contained incomplete data. Missing data were added to the Table 2 in the line myofascial pain (n=26, 100%). - according to Reviewer suggestions
6th: subgroups among HJS & concurrent TMD patients
Next, the myofascial pain group n=22 minus? should be differentiated from the internal derangement (ID) group with DDwR plus again the mixed group pain & ID derangement group t(so there are basically 3 different patient sub-collectives) to allow for sound nosologically based conclusions. Otherwise this means comparing apples and oranges (which comes back to the problem of small numbers problem address in line313)
In this context, with a rate of approximatively 10% among TMD, larger collectives could be realized e.g. by a multicentre design.
Due to the small sample size, it was decided to collectively analyze mixed types of TMD. Among 26 participants 16 of them had disc displacement with reduction plus myofascial pain and the remaining 10 patients had only myogenic TMD. Information added in the lines 212-214. - according to Reviewer suggestions
7th: assessment of MMO as parameter for effectiveness:
The MMO parameter is highly questionable: does it really make sense to even reinforce mobility of an already hypermobile joint and use this as an indicator for effectiveness of physiotherapy? MMO is rather a surrogate parameter for reduction of muscle pain, so the authors should focus on myofascial pain as target parameter in HJS patients (if present, cf. above), at least this point needs to be addressed in the discussion part.
The examination of MMO was included to provide an overall evaluation of the effect of physiotherapy on the mobility of the TMJs. The goal of the therapy was not to increase the TMJs range of motion, but to improve its coordination through strengthening exercises. The main problem of most patients was myofascial pain, the elimination of pain was one of the goals of the therapy. The explanation of this issue is discussed further in the results line: 214-217.
8th: conclusion part is highly speculative:
a) conclusion can occur with (generalized) HJS (without control group of both a normal population and a non HJS control group) this massage is either trivial (of course they can occur) or unfounded by supplied data;
b) related to increased ROM (cannot be stated without control group and respective statistical evaluation, so conclusion remains speculative)
c) myofascial pain (cf. above, plus no subgroup analysis performed, therefore not based on presented data and thus speculative)
d) articular related dysfunction (i.e. internal derangement Wilkes I & II) : cf. above, not supported due to lack of a control group
e) coordination disturbances: mentioned, but no analysis done, this not supported by data, speculative
f) physiotherapy is effective in reducing pain (yes, but no control without HJS, so less meaningful) and improving the coordination of the mandible (no data supplied, therefore speculative)
a)b)c) Studies on the topic are limited and the theoretical foundation is poor, however we considered this limitation as an opportunity to identify new paradigm in physiotherapy and prevention for patients with HJS and highlight the need for further development in this area of study and we plan further research with the control group- according to Reviewer suggestions
- d) articular related dysfunction (i.e. internal derangement Wilkes I & II) were not supported because mean age of patient was 27 y.o. and MRI was not recommended for disc displacement with reduction in this cases.
- e) f) Coordination disturbances were assessed as the opening movement path using a movement diagram. The assessment was performed before and after intervention. Initially, the deviations during temporomandibular joint were obtained in 19 patients and at the end of intervention correction of the trajectory was obtained in 10 from 19 patients (52.6%). Missing data were added to the results (line 221-222). The lack of data results from a mistake when creating the tables for this article.- according to Reviewer suggestions
Some minor points to be addressed/clarified:
- sequentially assessed NPS is rather subjective evaluation tool – the authors should also state that it was in fact one examiner (with > 20 years of experience) or were there several examiners experienced > 20 years? Should be made clear. If several, interrater assessment would be required (IR variability etc.).
The examination and therapy of all patients was carried out by one and the same physical therapist (line 129-130).
- just to avoid any misunderstanding: the authors mention the term deviation assessed as more than 5 mm abduction from the midline; do the authors mean deflection instead or really deviation? (with the latter one of course making sense as there was no DDwoR in the collective)
The assessment of the 5 mm difference in abduction of TMJs concerned the range of the mouth opening. To avoid any misunderstandings, the term "abduction of TMJs" was changed to “maximum mouth opening”.- - according to Reviewer suggestions
Line 148-150:
“Increased maximum mouth opening was detected when the difference between tested maximum mouth opening and the individual physiological norm of mouth opening was more than 5 millimeters.”
The subjects had a deviation-type mandibular movement dysfunction – deviating from the midline to one side while mouth opening and back to the midline in the final range of mouth opening. More information added in the “Material and methods” line 152 and “Results” (line 202-205).
- last but not least the reviewer recommends to modify the title of the article to run: “The effectiveness of physical therapy in patients with generalized joint hypermobility and concurrent temporomandibular disorders - a cross sectional study”
The title has been changed (line 2-4). - according to Reviewer suggestions
Kindly thank You once again
With best wishes
Yours sincerely- the Authors

Reviewer 2 Report
This article ignores the strictly articular signs of TMJ hypermobility such as subluxation or episodes of open-mouth mandibular blockage.
Curiously, the rehabilitation does not use any of the exercises of mandibular retro imulses and of strengthening of the joint structures. Use of masticatory muscles stretching should be explained further because it seems paradoxical.
The measuremnt of the mouth opening by "3-fingers assessment" is not valid. The mouth openoing must be assessed in millimeters accordingto the standardDC/TMD and must take into account any over bite or gap between incisors.
What is the "abduction of TMJs"?
The musculature exam does not meet the criteria of the DC/TMD. given its subjective nature, it would have been preferable if it had been carried out by several practitioners.
how can increasing mouth opening be a goal of physiotheraywhen theses patients already suffer from hypermobility (48.4 mm before !!)
Is the TMJ clicking a noise at the start of opening which indicates reducible disc displacement or a finally opening noise which indicates an excessive travel of the condyle in front of the disc ?
Three weeks of physiotherapy seems a very short time
The words of the conclusion seem contradistory with what is said at the end of the discussion
Author Response
Response to Reviewer 2 Comments
Dear Reviewer,
We would like to thank you for reviewing our manuscript. We appreciate all comments and effort put into writing the review. We are thankful for your positive comments and constructive advice that has helped us improve the text. All the changes were introduced using the "Track Changes" function in Microsoft Word, so that changes are easily visible.
Kindly we thank You once again
With best wishes
Yours sincerely- the Authors
The followings are our point-by-point responses:
- This article ignores the strictly articular signs of TMJ hypermobility such as subluxation or episodes of open-mouth mandibular blockage.
The diagnosis of TMJs was performed according to the DC / TMD criteria by an experienced physiotherapist. None of the patients had been diagnosed with subluxation or open-mouth mandibular blockage during the initial examination, but disc displacement with reduction.
- Curiously, the rehabilitation does not use any of the exercises of mandibular retro imulses and of strengthening of the joint structures. Use of masticatory muscles stretching should be explained further because it seems paradoxical.
The relaxation techniques were aimed at reducing myofascial pain, not lengthening the structure of the muscle, focusing on the relaxation of the fascia. Mandibular movement coordination exercises were used, included strengthening exercises and isometric exercises. The detailed description of the intervention and goals of the exercises (line 126-130).
- The measurement of the mouth opening by "3-fingers assessment" is not valid. The mouth openoing must be assessed in millimeters accordingto the standardDC/TMD and must take into account any over bite or gap between incisors.
The examination of the mouth opening movement was performed in accordance with the DC/ TMD-based methodology and considering malocclusion and was assessed in milimeters. However, none of the patients included in the study had malocclusions such as cross or open bite.
"3-fingers assessment" was an additional measurement used to relate the maximum mouth opening to physiological norms and to demonstrate increased range of motion (overmobility of TMJ in HJS.
Additionally, the results were assessed according to the DC / TMD standards. Interpretation of results according to DC / TMD motion ranges was added in the table 2, line “maximum mouth opening”.
- What is the "abduction of TMJs"?
To avoid nomenclature confusion, the term "abduction of TMJs" has been changed to "maximum mouth opening" in the text. (lines: 135, 149-150, 200)- )- Corrected according to Reviewer suggestions
- The musculature exam does not meet the criteria of the DC/TMD. given its subjective nature, it would have been preferable if it had been carried out by several practitioners.
The study was conducted by one physiotherapist but with > 20 years of experience in physiotherapy for TMD, if there were several examiners interrater assessment would be required (IR variability etc.).
- how can increasing mouth opening be a goal of physiotherapy when theses patients already suffer from hypermobility (48.4 mm before !!)
The goal of physiotherapy was not to increase the TMJs range of motion. There was no such purpose in the description of the intervention (line 123-125). The 'mouth opening' variable was included in the analysis of the results only to assess the effect of the exercises on range of motion. The direct aim of which was not to increase mobility but to improve coordination of movement and joints stability.
Perhaps the increase in range of motion (Results, Table 3.) was due to the relaxation of tense muscles. The therapy did not use exercises aimed at increasing the range of motion, but relaxation exercises and trigger point therapy aimed to tense and painful muscles. More information added in the lines 214-217. - Corrected according to Reviewer suggestions
- Is the TMJ clicking a noise at the start of opening which indicates reducible disc displacement or a finally opening noise which indicates an excessive travel of the condyle in front of the disc?
16 patients were diagnosed with disc displacement with reduction ; the TMJ clicking a noise at the start of opening which indicated disc displacement with reduction, it was not opening noise which could indicate an excessive travel of the condyle in front (anteriorly).
- Three weeks of physiotherapy seems a very short time
We realize that the duration of the intervention was short. These limitations resulted from reasons beyond the control of the authors, related to finances and personal issues. Added this conclusion to the study limitations and recommendations for further research.
- The words of the conclusion seem contradistory with what is said at the end of the discussion
Studies on the topic are limited and the theoretical foundation is poor, however we considered this limitation as an opportunity to identify new paradigm in physiotherapy and prevention for patients with HJS and highlight the need for further development in this area of study and we plan further research with the control group and a long term of follow-up assessment.
Kindly thank You once again
With best wishes
Yours sincerely- the Authors

Round 2
Reviewer 1 Report
The authors addressed and corrected some of the major criticisms, but did not supply a control group as previously recomended.
1st: lack of control group
authors# rebuttal:
The study was carried out as an analysis of the medical records of patients who underwent physiotherapy due to dysfunctions in temporomandibular joints. Among them, only patients with TMD and HJS were selected. The lack of a control group was due to the large variation in the remaining data and missing data. It is planned to extend this study to a control group with a repeat of the study protocol. We added the limitations and pointed out strengths - Corrected according to Reviewer suggestions (lines: 330-332).” The third limitation is the lack of a control group due to the retrospective nature of the study. Future studies should include a control group and a long term of follow-up assessment”
Comment: Apparently there is a misunderstanding regarding the feasilbility and necessity of the requested control group: the authors declare that they analysed patients’ records who underwent physiotherapy due to TMJ dysfunctions. They only selected patients with TMD and HJS; so a control group means that they should now randomly select from the more than 250 remaining records a comparable group of patients with TMD but without HJS; from the reviewer’s standpoint it is not plausible why it should not be possible to supply such a control group and why this should be postponed into the future, instead. The authors give the information that they overall assessed over 300 (!) patients –therefore, it is hardy plausible that just the HJS patients had overall full records but that the remaining 250 + did not. Selection criteria: myogenic TMD, DDwR, no HJS
To sum up: As this is in fact still a major drawback of the article, this should be amended for the present publication.
3rd: DDwR and HJS
Please also note that DDwR may be attributed to HJS, but there are no data given in the article supporting this conclusion; at least not unless data of a representative control group are supplied allowing for a comparison; thus, this potentially interesting information remains speculative and is not supported by data (solution Table 1: should contain the data for the overall collective without HJS)
The results in the Table 2 and in the lines 212-214 shows that among study group there were 16 patients with disc displacement with reduction and co-occurred myofascial pain
Comment: that’s exactly why a control group and respective statistics are mandatory, otherwise the conclusion cannot be drawn (although it is highly plausible and it is also highly likely that statistics will reveal significant differences if (!) compared with a randomly selected control group; but without a control this remains speculative, then
Table 1. Baseline characteristics of the patients with HJS- only patients with TMD and HJS were selected. The lack of a control group was due to the large variation in the remaining data and missing data. It is planned to extend this study to a control group with a repeat of the study protocol- it was pointed in the discussion.
From the reviewer’s standpoint, it is not acceptable not to supply a control group already at the present stage, cf. arguments above
To sum up: As previously strongly (i.e mandatory!) recommended, the authors should supply a control group, then perform the respective statistics and then the paper will gain better quality, reach a higher LoE and will certainly be worth publishing, then.
Author Response
Response to Reviewer Comments
Dear Reviewer,
We would like to thank you for reviewing our manuscript. We appreciate all comments and effort put into writing the review. We are thankful for your positive comments and constructive advice that has helped us improve the text. All the changes were introduced using the "Track Changes" function in Microsoft Word, so that changes are easily visible.
The followings are our point-by-point responses:
“The authors addressed and corrected some of the major criticisms, but did not supply a control group as previously recomended.
1st: lack of control group
authors# rebuttal:
The study was carried out as an analysis of the medical records of patients who underwent physiotherapy due to dysfunctions in temporomandibular joints. Among them, only patients with TMD and HJS were selected. The lack of a control group was due to the large variation in the remaining data and missing data. It is planned to extend this study to a control group with a repeat of the study protocol. We added the limitations and pointed out strengths - Corrected according to Reviewer suggestions (lines: 330-332).” The third limitation is the lack of a control group due to the retrospective nature of the study. Future studies should include a control group and a long term of follow-up assessment”
Comment: Apparently there is a misunderstanding regarding the feasilbility and necessity of the requested control group: the authors declare that they analysed patients’ records who underwent physiotherapy due to TMJ dysfunctions. They only selected patients with TMD and HJS; so a control group means that they should now randomly select from the more than 250 remaining records a comparable group of patients with TMD but without HJS; from the reviewer’s standpoint it is not plausible why it should not be possible to supply such a control group and why this should be postponed into the future, instead. The authors give the information that they overall assessed over 300 (!) patients –therefore, it is hardy plausible that just the HJS patients had overall full records but that the remaining 250 + did not. Selection criteria: myogenic TMD, DDwR, no HJS
To sum up: As this is in fact still a major drawback of the article, this should be amended for the present publication.”
Response: A control group was added. The inclusion criteria for control group were myogenic TMD and combined myogenic TMD with disc displacement with reduction to obtain the most homogeneous group. Detailed description was added in “Material and Methods” and “Results”. Therefore, information in the abstract and conclusions were changed.
“3rd: DDwR and HJS
Please also note that DDwR may be attributed to HJS, but there are no data given in the article supporting this conclusion; at least not unless data of a representative control group are supplied allowing for a comparison; thus, this potentially interesting information remains speculative and is not supported by data (solution Table 1: should contain the data for the overall collective without HJS)
The results in the Table 2 and in the lines 212-214 shows that among study group there were 16 patients with disc displacement with reduction and co-occurred myofascial pain
Comment: that’s exactly why a control group and respective statistics are mandatory, otherwise the conclusion cannot be drawn (although it is highly plausible and it is also highly likely that statistics will reveal significant differences if (!) compared with a randomly selected control group; but without a control this remains speculative, then
Table 1. Baseline characteristics of the patients with HJS- only patients with TMD and HJS were selected. The lack of a control group was due to the large variation in the remaining data and missing data. It is planned to extend this study to a control group with a repeat of the study protocol- it was pointed in the discussion.
From the reviewer’s standpoint, it is not acceptable not to supply a control group already at the present stage, cf. arguments above
To sum up: As previously strongly (i.e mandatory!) recommended, the authors should supply a control group, then perform the respective statistics and then the paper will gain better quality, reach a higher LoE and will certainly be worth publishing, then.”
Response: A control group was added. The comparison of prevalence disc displacement with reduction between “HJS group” and “group without HJS” was added in the Table 2. Detailed information was added also in Results.
